# Automatic Classification of Photos by Tourist Attractions Using Deep Learning Model and Image Feature Vector Clustering

**Jiyeon Kim and Youngok Kang ***

Department of Social Studies, Ewha Womans University, Seoul 03760, Korea; jy.kim20@ewhain.net
* Correspondence: ykang@ewha.ac.kr

**Abstract:** With the rise of social media platforms, tourists tend to share their experiences in the form of texts, photos, and videos on social media. These user-generated contents (UGC) play an important role in shaping tourism destination images (TDI) and directly affect the decision-making process of tourists. Among UGCs, photos represent tourists' visual preferences for a specific area. Paying attention to the value of photos, several studies have attempted to analyze them using deep learning technology. However, the research methods that analyze tourism photos using recent deep learning technology have a limitation in that they cannot properly classify unique photos appearing in specific tourist attractions with predetermined photo categories such as Places365 or ImageNet dataset or it takes a lot of time and effort to build a separate training dataset to train the model and to generate a tourism photo classification category according to a specific tourist destination. The purpose of this study is to propose a method of automatically classifying tourist photos by tourist attractions by applying the methods of the image feature vector clustering and the deep learning model. To this end, first, we collected photos attached to reviews posted by foreign tourists on TripAdvisor. Second, we embedded individual images as 512-dimensional feature vectors using the VGG16 network pre-trained with Places365 and reduced them to two dimensions with t-SNE(t-Distributed Stochastic Neighbor Embedding). Then, clusters were extracted through HDBSCAN(Hierarchical Clustering and Density-Based Spatial Clustering of Applications with Noise) analysis and set as a regional image category. Finally, the Siamese Network was applied to remove noise photos within the cluster and classify photos according to the category. In addition, this study attempts to confirm the validity of the proposed method by applying it to two representative tourist attractions such as 'Gyeongbokgung Palace' and 'Insadong' in Seoul. As a result, it was possible to identify which visual elements of tourist attractions are attractive to tourists. This method has the advantages in that it is not necessary to create a classification category in advance, it is possible to flexibly extract categories for each tourist destination, and it is able to improve classification performance even with a rather small volume of a dataset.

**Keywords:** image feature vector; clustering; Siamese Network; automatic classification of tourist photos; deep learning model

## 1. Introduction

Recently, as anyone can access social media platforms anytime, anywhere using mobile devices, a large volume of texts and photos have been shared on the web to communicate with others. People are freely expressing their thoughts and feelings through text and photos on social media platforms. Along with this trend, the way in which tourists get information related to travel attractions and share their experiences is also changing. More and more tourists share their experiences in the form of texts, photos, and videos on social media, which serves as an information source for potential tourists [1]. Data posted on social network services (SNS) is steadily receiving social attention in that it is user-generated content (UGC). The tourism industry is also paying attention to UGC data to identify new tourism trends and analyze the image of tourist attractions perceived by tourists [2]. In

particular, the image of tourist attractions plays an important role when people select their tourist destination and destination marketing organizations (DMO) perform tourism marketing [3–6].

In the past, DMOs have played a leading role in shaping the image of tourist destinations. However, due to the popularization of social media platforms in recent years, it has been recognized that the image of tourist attractions is formed by both UGC and the contents created by DMOs [7]. Among UGCs, a photo plays an important role in forming the image of tourist attractions in that it visually reproduces the places [8]. A photo reflects the mental image of the physical elements experienced by the photographers. In addition, a photo is a record of a moment to express a mental image of a place in a visual form [9]. Therefore, since these photos contain tourists' visual preferences for a specific area, they can reflect actual tourists' preferences more directly than a few experts [10]. In addition, potential tourists tend to visit tourist sites that have been exposed to them and take pictures of visual images that have been projected on them [11].

Paying attention to the value of photos, more and more studies have attempted to analyze photos on SNS taken by tourists and uncover attractive factors that contribute to the formation of the image of a tourist destination [1,4,6,12–14]. However, due to the limitations in technologies, studies on tourism destination images (TDI) using UGC photos encounter challenges in terms of both the volume of data and the interpretation of results. The most widely used method is a manual analysis where researchers observe their collected photos and manually classify them into specific categories. Since this methodology is a labor-intensive process, there is a limit to the number of photos that can be analyzed, which makes it difficult to comprehensively analyze tourist attractions.

As computer vision technologies have developed, several studies have identified TDI from a number of SNS photos using deep learning methods [15–20]. However, they have limitations in classifying photos that represent unique characteristics of tourist attractions. They use predetermined photo categories such as Places365 or ImageNet which are designed for general purposes, so they are not appropriate for identifying the uniqueness of individual attractions. To overcome these limitations, Kang et al. and Yoon and Kang analyzed the images by generating a tourism photo classification category according to a specific area and training the model with training datasets for each category [21,22].

Although these existing studies have presented valuable results with the combination of UGC photos and a deep learning model to extract tourism destination images, studies are still in their infancy. In particular, studies on extracting distinctive characteristics of individual tourism attractions are limited. They have focused on analyzing the TDIs of a nation or a city rather than individual tourist attractions. While they also partially explore individual tourist' attractions included in the region, their categories for photo classification which are based on a national scale or city scale are not appropriate for figuring out the unique properties of individual tourism attractions.

To solve this problem, we propose a method for automatically building categories for photo classification using clustering and a Siamese network. This reduces the burden on the process of creating categories corresponding to each tourism attraction. In addition, the clustering methodology provides the advantage of establishing categories based on a data-driven manner. Our framework consists of the following four parts. First, we collected TripAdvisor photos in reviews posted by foreign tourists in Seoul. Second, we embedded individual images as 512-dimensional vectors using a VGG16 network pretrained with Places365 and reduced these vectors to two dimensions with t-SNE. Third, to create a category based on visual content that frequently appears in photos taken by tourists, clusters were extracted through HDBSCAN analysis and they were set an image category of an attraction. Finally, a Siamese Network was applied to remove noise photos within the cluster and classify photos according to the category.

## 2. Literature Review

### 2.1. Analysis of Tourist Attractions Using UGC Photos

With the popularization of mobile devices and the rise of social media platforms, the images of tourist attractions tend to be formed through photos and narratives shared online. The shared images of tourist attractions are continuously perceived and reproduced from person to person [12]. As content posted on social media platforms are exposed to many people, they tend to travel to destinations or attractions that frequently appear on the SNS. These visual images allow DMOs to get insights into tourist behaviors and perceptions for marketing. Compared to existing marketing tools, this type of marketing is recognized as an effective tool that quickly affects the decision-making process of a tourist while reducing costs [13]. Paying attention to the value of such photos, more and more studies are attempting to analyze photos taken by tourists and uncover attractive factors of tourist destinations. Before the rise of the deep learning method in a tourism context, the predominant method in the analysis of photos is to directly observe them one by one, which is a manual manner. Agustí et al. and Dinh identified the process of forming a tourism image in a specific area through the analysis of these photos [1,12]. Stepchenkova et al. analyzed the difference between the image generated by tourists and the image projected by DMOs [14]. In the case of direct visual observation, which requires a labor-intensive process, it is difficult to comprehensively analyze tourist destinations because there is a limit to the volume of photos that can be analyzed. In addition, there is another limitation that the research results may be dependent on researchers.

With the rapid development of computer vision and image processing technologies in recent years, several studies that analyze a large volume of photos using deep learning models are emerging in the tourism field. Most of the studies have applied a convolutional neural network (CNN) developed to solve the image classification problem. These studies have classified photos according to specific categories and uncovered tourists' perceptions of specific areas based on their classification ratios.

Most of the studies analyzed tourist photos using a pre-trained model with Places365 which is a dataset specialized in place classification problems [15–20]. However, according to Kim et al., when using a pre-trained model, there is a problem of misclassifying unique objects or scenes that appear in photos of local tourist attractions [17]. To overcome this limitation, other studies that have transferred models to training datasets specialized in research areas have emerged. Kang et al. and Yoon and Kang analyzed tourism images in a specific area through transfer learning of a deep learning model after constructing categories and datasets specialized in that area without using a pre-trained model [21,22]. These studies, which have used categories to classify tourism images, have limitations in that it is difficult to properly identify the features of local tourist attractions and it takes a lot of time and effort to build training datasets manually.

### 2.2. Application of Deep Learning-based Image Embedding and Clustering

Clustering, one of the representative unsupervised learning methodologies, enables us to discover hidden patterns and structures in data. In order to perform clustering analysis, a process of extracting features of image and converting them into vectors is required. Before the rise of the deep learning method, image embedding algorithms that extract fixed feature points from the image, such as scale-invariant feature transform (SIFT), speeded up robust feature (SURF), and binary robust independent elementary features (BRIEF), have been used in this process [23]. With the rapid development of computer vision technology, CNN-based auto-encoders and embedding model based on the CNN network have been widely used. The latter methods transform images into vectors through feature maps in the CNN network pre-trained on specific datasets such as Places365 or ImageNet [24]. Image clustering provides a way for discovering hidden structures or patterns of data in various fields.

Tapaswi et al. embedded faces based on a deep learning model to determine the number of characters appearing in the video, then identified the number of clusters through

hierarchical cluster analysis and derived the number of characters [25]. Gu et al. identified New York's fashion trends by embedding street fashion images and applying agglomerative hierarchical clustering by year [26]. Castellano and Vessio converted the artwork image into a feature vector with the DenseNet121 network, applied K-means clustering and auto-encoder to find the cluster, and analyzed the painting style of the artwork through cluster results [27].

### 2.3. Application of Siamese Network with Image Embedding

Siamese Network is particularly used in the research of medical care, palm print, face recognition, object tracking, etc. where it is difficult to obtain a large volume of data. Siamese Network has been applied to solve these problems. When two or more input images are given, Siamese Network learns the similarity between them and expresses it as a numerical distance. That is, if the input images are similar to each other, the distance is close, and if the input images are different, the distance becomes far. The same principle can be applied to image classification in various fields.

Schroff et al. developed a FaceNet model learned with triplet loss for face recognition based on the Siamese Network structure [28]. The FaceNet model embeds an input face image in 128 dimensions, and then distinguishes between photos of a person's face and photos that do not, through the distance between the embedded vectors. Zhong et al. developed a palm print recognition model using Siamese Network based on the VGG16 network [29]. In this study, they used Siamese Network to convert a long text image given as input data into a 500-dimensional vector and compared the distances between the two long text image vectors to determine whether they were identical. Mehmood et al. developed a model for early detection of Alzheimer's by utilizing the VGG-16 network-based Siamese Network [30]. They trained a model using MRI datasets classified into four types according to the severity of Alzheimer's and classified the progression of Alzheimer's by comparing the distance between embeddings. Bertinetto et al. developed an object tracking model based on Siamese Network [31]. They embedded the image containing the object to be tracked and the image to find the object into Siamese Network and identified the location of a specific object in the image by comparing the similarity between embeddings. As such, these studies are widely used, especially in cases where it is difficult to secure sufficient datasets such as long palm print, face recognition, and disease diagnosis. Even in the tourism field, if the scale of the research target is narrowed down to a specific tourist area, it may be difficult to secure sufficient data. Therefore, this study also intends to use the Siamese Network model to improve classification performance with a rather small volume dataset.

## 3. Materials and Methods

### 3.1. Research Process

The analysis process of this study is shown in Figure 1. First, we extracted the photos posted on TripAdvisor and then selected the photos posted by foreign tourists, excluding Koreans using the publisher's country of origin and the language used in writing the review. Second, we embedded individual images as 512-dimensional feature vectors using a VGG16 network pre-trained on the Places365 dataset. Third, we reduced each vector to two-dimension with t-SNE, employed HDBSCAN to cluster these photos, and set this as an image category. Fourth, we used Siamese Network to remove noise images not included in categories and classify photos according to the category.

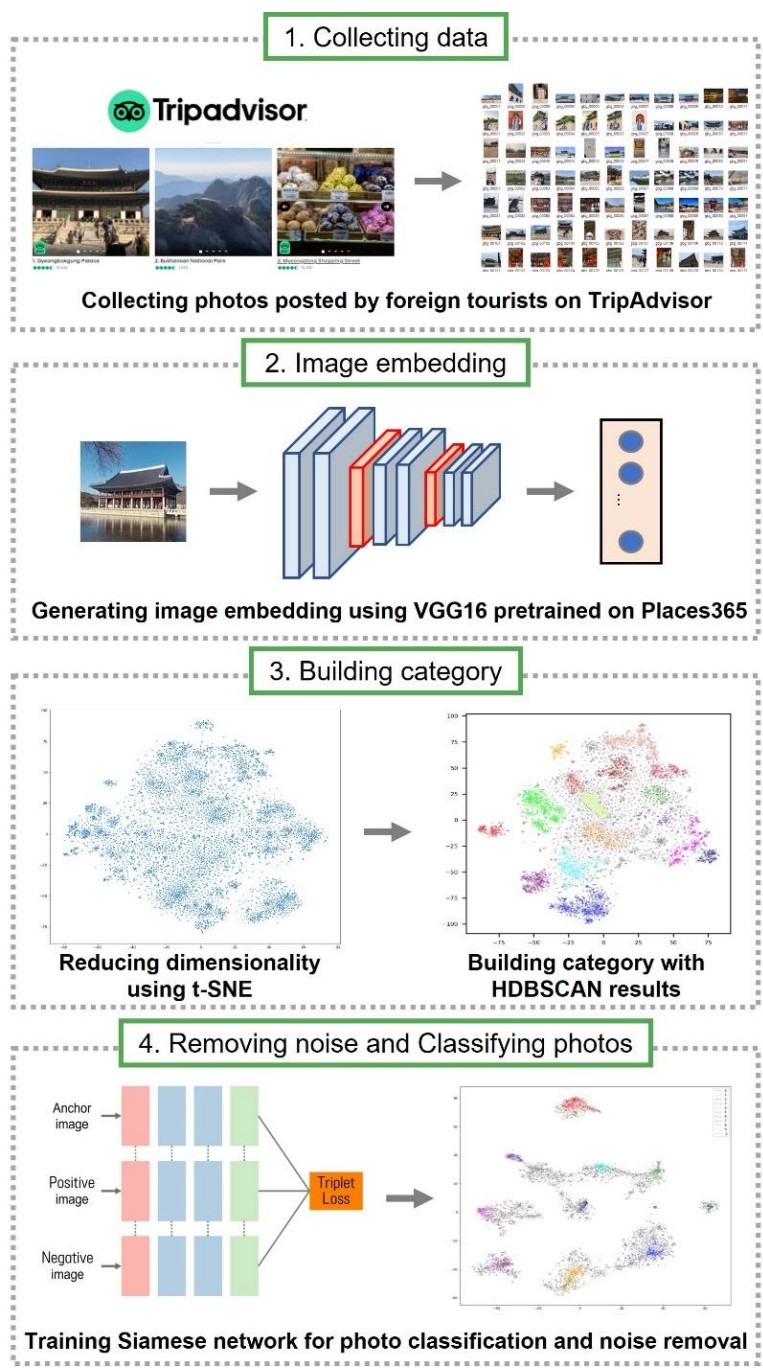

**Figure 1.** Research Process.

### 3.2. Collecting Data

TripAdvisor (www.tripadvisor.com, accessed on 19 September 2021) is the world's largest travel information platform with more than 260 million monthly users [32]. TripAdvisor provides reviews written by visitors to tourist attractions, hotels, and restaurants around the world. When you log on to TripAdvisor and search for a country or city, you can find information about popular tourist attractions, accommodations, restaurants, and activities in that area. If people search for Seoul on TripAdvisor, they can find a list of famous tourist attractions in Seoul under the heading 'Top Attractions in Seoul'. If people click on each tourist attraction, they can check the reviews posted by tourists who have visited the tourist attractions. In each review, they can identify the user nickname, region of origin, the number of posts, star rating, review title, date of visit, type of visit, textual

body, attached photos, and date of creation. In this study, we collected reviews of 'Gyeong-bokgung Palace' and 'Insadong' using Python, and collected photos attached to the review, date of visit, and nationality data of the publisher. 'Gyeongbokgung Palace' is a palace of the Joseon Dynasty located in the city center and is one of the most visited places by foreign tourists. 'Insadong', a distinctive area with a mixture of galleries, traditional restaurants, modern buildings, and shopping streets, is also a popular place for foreign tourists. In this study, we only selected the photos posted by foreign tourists using language filtering and the visitors' origin information. As the number of inbound tourists dropped sharply in 2020 due to COVID-19, all data before 2020 were used.

### 3.3. Image Embedding

To analyze image data using deep learning, it is necessary to map the image into an embedding space. In this case, when a vector is generated by arranging pixel values of an original image in a row, it is difficult to determine the similarity between images through distance measurement because the vector does not reflect the case where the same visual pattern exists at different locations in the picture. As an alternative to this, a method of extracting a visual pattern of an image and embedding it as a vector can be used.

In this study, we utilized a CNN-based embedding model where vectors were generated by reflecting the visual content of the photo. Therefore, vectors having similar visual contents are located close to each other in the embedding space and vice versa. The close distance between vectors means that the visual contents of the original images are similar. The CNN-based image classification model is largely divided into two parts: one is to learn the features of images and the other is to classify images based on the features. The former consists of a convolutional layer, an activation function, and a pooling layer, while the latter consists of a fully connected layer and a softmax layer. Since there is no need for the latter part in the embedding process, we replaced the fully connected layer at the top of the CNN model with a Global Max pooling layer. In this study, we employed a VGG16 network pre-trained on Places365 dataset for embedding. Places365 is a benchmark dataset created by extracting 365 categories from Place datasets consisting of a total of 10 million photos [33]. Since this study tries to analyze photos taken at tourist attractions, a pre-trained model with Places365 was used.

### 3.4. Dimension Reduction and Clustering

The major functions of this process are to extract visual contents that frequently appear in photos taken by tourists and to utilize them as a category. First, we reduced the 512-dimension to 2-dimension with t-SNE. Second, we employed the HDBSCAN clustering algorithm to cluster these embeddings. The clustering results showed tourists' visual preference for a tourism attraction that forms TDI.

The t-SNE is one of the nonlinear methods designed to reduce high-dimensional data to two or three dimensions based on probability distribution and visualize it [34]. This method was developed by supplementing the problems of Stochastic Neighbor Embedding [35] and is focused on maintaining a local structure when reducing dimensions. Here, maintaining the local structure means reducing the data so that the relationship can be kept even after the points that are close to each other in the high dimension are projected to the low dimension. In Equation (1), $p_{ij}$ represents the probability that data points $x_i$ and $x_j$ existing in a high dimension are neighboring to each other. In Equation (2), $q_{ij}$ represents the probability that $x_i$ and $x_j$, which are low-dimensional points corresponding to $y_i$ and $y_j$, are adjacent to each other. The cost function of t-SNE is calculated by Kullback-Leibler divergence, a function that calculates the difference between probability distributions of both a high and a low dimension in Equation (3) [36].

$$p_{ij} = \frac{exp(- \parallel x_i - x_j \parallel^2 /2\sigma^2)}{\sum_{k \neq l} exp(- \parallel x_k - x_l \parallel^2 /2\sigma^2)} \tag{1}$$

$$q_{ij} = \frac{exp(-\|y_i - y_j\|^2)}{\sum_{k \neq l} exp(-\|y_i - y_j\|^2)} \tag{2}$$

$$C = KL(P \| Q) = \sum_i \sum_j p_{ij} \log \frac{p_{ij}}{q_{ij}} \tag{3}$$

$p_{ij}$: joint probability that $i$ and $j$ are neighbors in a high dimension
$q_{ij}$: joint probability that $i$ and $j$ are neighbors in a low dimension
$x_i, x_j$: high-dimensional data points
$y_i, y_j$: low-dimensional data points counterparts of the $x_i$ and $x_j$

We employed the HDBSCAN algorithm to identify clusters in embedding space and utilize them as a category. HDBSCAN has evolved from density-based spatial clustering with noise (DBSCAN), a density-based clustering algorithm [37]. DBSCAN finds a cluster of points over a certain density in the entire data point space [38]. Here, the certain density is defined as the value of Eps indicating the radius and $m_{pts}$, which is the minimum number of data points included in the Eps. DBSCAN has two drawbacks, the first being sensitive to parameters, and the second being that it cannot find clusters with different densities because it sets thresholds for density. HDBSCAN is an algorithm that compensates for the shortcomings of DBSCAN, and adds the concept of hierarchical clustering to DBSCAN. Since HDBSCAN finds clusters by defining only the minimum amount of data without using a fixed Eps value, it can extract various clusters with different density values.

*3.5. Removing Noise Data and Classifying Photos*

It may seem that clustering results could replace photo classification in that clusters are made up of similar visual contents. However, since HDBSCAN works based on density, the accuracy of clustering tends to be low at the edge of a cluster that has a relatively low density than the core. This is responsible for two problems. First, noise photos that are not related to the cluster may be included. Second, noise points located around the boundary of the cluster may not actually be noise.

To address these problems, we implemented a Siamese network to classify photos trained with our own dataset made up of photos taken in the research area. Siamese network consists of more than two identical subnetworks capable of learning patterns from input vectors [39]. The outputs generated through a Siamese network reflects the similarity between images. Although the model receives different photos as inputs, the weights in subnetworks are updated equally because they are combined by the loss function. This weight fixation means that visually similar images are located close to each other and vice versa.

In this study, triplet loss was used as a cost function for training the Siamese network, and semi-hard was used among triplet mining methods. The triplet loss function receives three types of input data: anchor, positive, and negative. There are three ways to construct input data: easy triplets, hard triplets, and semi-hard triplets. Schroff et al. revealed that the model trained using the semi-hard triplets' method is superior among them [28]. Figure 2 shows the architecture of the model used in the study.

Siamese network learns to position the images with similar visual content close to each other in vector space and vice versa. Based on this principle, photos can be classified, and noises can be detected. For this, a target set, a noise set, and a reference set are needed. The target set is a set of target photos to be classified and labelled according to category items. A noise set is a set of noise photos that do not belong to the category. The reference set consists of sample images of each category. Test sets and reference sets were made to consist of a similar number of photos for each category. The noise set was made with a similar number of photos to the test set. Photo classification and noise removal are performed through the following four steps: First, the distance between a target photo and the photo belonging to the reference set is calculated respectively. Second, the prediction label of the target image is assigned as a label of a reference photo having a minimum distance in Equation (4). These processes are repeated for all target images. Third, the distance between a noise photo and

the photo belonging to the reference set is calculated, respectively. This step is repeated for all photos in the noise set. As a result, noise photos can be deleted by setting a threshold for the minimum distance. Fourth, the accuracy of prediction was evaluated as the minimum distance threshold was changed using the ROC curve. The threshold of the point showing the best accuracy was selected. The ROC curve is a graph showing how the performance of the classification system changes according to various thresholds. In this study *X* and *Y* axes of the ROC curve are True Positive Rate (TPR) and False Positive Rate (FPR). TPR is the percentage of cases where the true label matches the predicted label in the target set in Equation (5). FPR is the percentage of cases where noise is not classified as noise in Equation (6). The optimal threshold is the value of the point farthest from Y = X among points on the ROC curve in Figure 3.

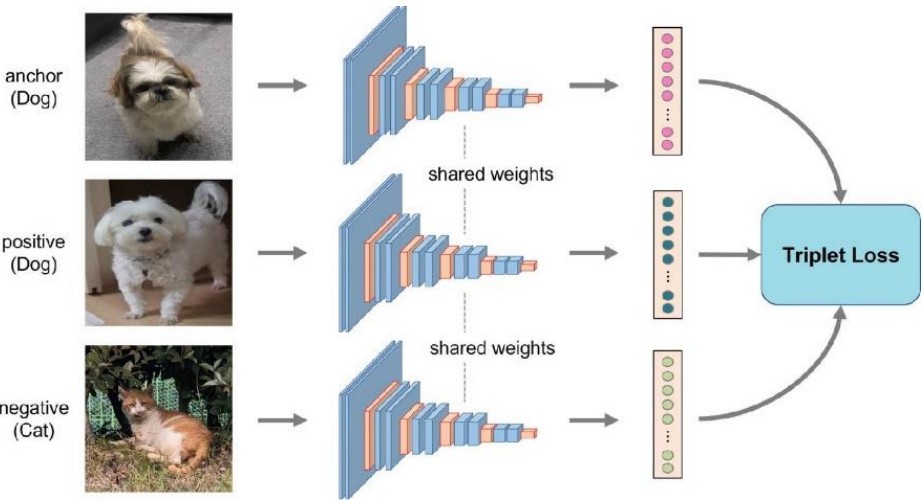

**Figure 2.** Model architecture based on semi-hard triplet Siamese Network.

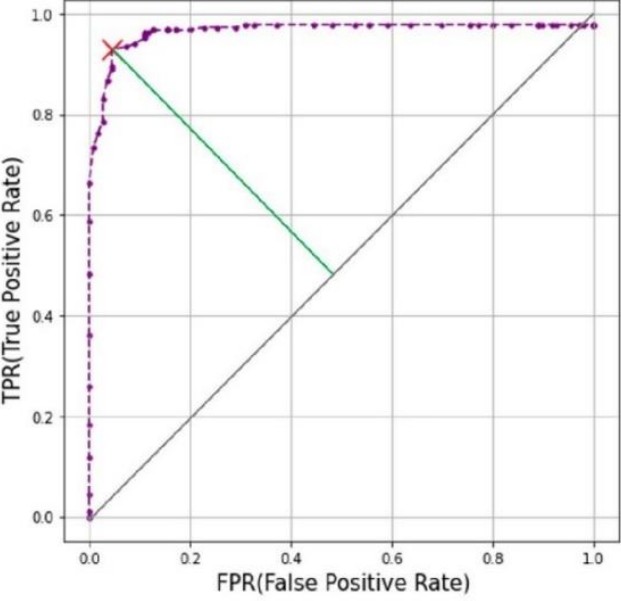

**Figure 3.** Example of ROC Curve.

$$l(t) = \underset{i}{\mathrm{argmin}}\big(d\big(t, S_{i,\,j}\big)\big) \tag{4}$$

$l(t)$: predicted label on a target photo
$t$: image embedding of a target image
$S_{i,j}$: Embedding of the *j*th sample image of category item *i*.

$d(x, y)$: Euclidean distance function between $x$ and $y$

$$TPR = \frac{n(TL(i) = PL(i),\ i \in testset)}{n(testset)} \tag{5}$$

$$FPR = \frac{n(TL(i) \neq PL(i),\ i \in noiseset)}{n(noiseset)} \tag{6}$$

$TL(i)$: true label of a photo
$PL(i)$: predicted label of a photo

## 4. Results

### 4.1. Gyeongbokgung Palace

A total of 9940 photos were collected in 10,655 reviews registered on TripAdvisor on the 'Gyeongbokgung Palace' page. Out of a total of 9940 photos, we selected 8188 photos except for 715 reviews written in Korean. A VGG16 model pre-trained with Places365 embedded each photo in a 512-dimensional vector, and t-SNE reduced the vectors to two dimensions. We implemented HDBSCAN to cluster these vectors into several groups. The result is shown in Figure 4, and the number of photos for each cluster is shown in Table 1. Sixteen clusters were generated and 3824 points were classified as noise.

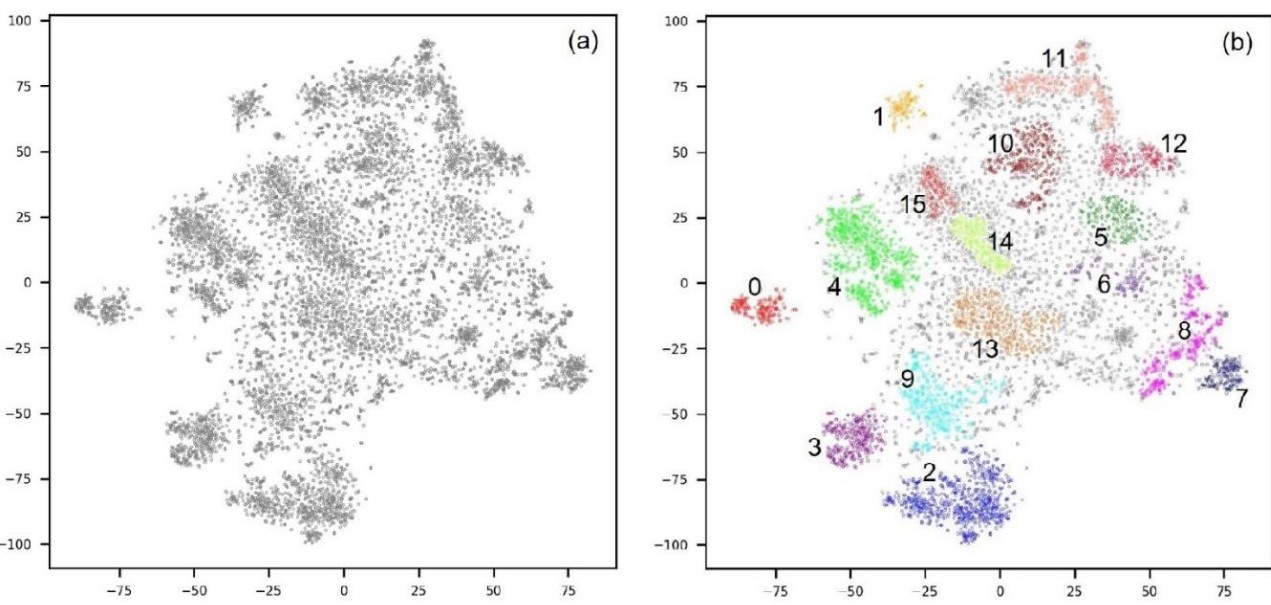

**Figure 4.** Gyeongbokgung Palace photos: (**a**) two-dimensional visualization; (**b**) result of HDBSCAN.

**Table 1.** Number of photos for each cluster generated as a result of HDBSCAN in Gyeongbokgung Palace.

| Cluster | Number of Photos | Cluster | Number of Photos |
|---------|-----------------|---------|-----------------|
| 0 | 189 | 8 | 412 |
| 1 | 124 | 9 | 381 |
| 2 | 715 | 10 | 380 |
| 3 | 358 | 11 | 391 |
| 4 | 737 | 12 | 239 |
| 5 | 199 | 13 | 456 |
| 6 | 146 | 14 | 235 |
| 7 | 173 | 15 | 183 |

After inspecting the photos in each cluster, we took two actions. First, if there were more than two clusters that contained the same visual contents, we integrated them into one. Second, if there was no similarity between the pictures that formed a cluster, that cluster was deleted because it is difficult to consider them as a meaningful cluster. Clusters 7

and 8 were integrated into one cluster because both were composed of photos of 'Throne'. Clusters 10 and 12 were also combined into one cluster because both consisted of photos of the 'Gate guard changing ceremony'. Clusters 14 and 15 were also integrated into one cluster because both were composed of photos of 'Heungnyemun gate' in the same way. On the other hand, clusters 11, 12, and 13 were not used to create a category because each cluster was made up of different photos. As a result, 10 categories were created as follows: 'Gyeonghoeru Pavilion', 'Geunjeongjeon Hall', 'Heungnyemun Gate', 'Gwanghwamun Gate', 'Hangwomen Pavilion', 'National Folk Museum', 'Throne', 'Hanbok (traditional dress of Korea)', 'Gate guard changing ceremony', and 'Tree'.

Siamese network enables us to classify photos according to the previously generated category and remove noise photos. For this purpose, we trained a Siamese network based on the VGG16 network using Gyeongbokgung Palace's photo dataset. In this process, the training dataset was composed of the photos in each cluster except for those that were incorrectly included in clusters. Table 2 shows the number of photos included in the training dataset for each category.

**Table 2.** Number of training photos by category in Gyeongbokgung Palace.

| Category | Number of Training Photos |
|---|---|
| Gyeonghoeru Pavilion | 365 |
| Geunjeongjeon Hall | 342 |
| Heungnyemun Gate | 270 |
| Gwanghwamun Gate | 196 |
| Hangwonjeong Pavilion | 174 |
| National Folk Museum | 101 |
| Throne | 156 |
| Hanbok(traditional dress) | 145 |
| Gate guard changing ceremony | 173 |
| Tree | 151 |

To improve the ability of the model for pattern extraction through nonlinearity, two convolutional layers were added to the basic structure of the VGG16 network. The model used in this process had weights pre-trained on Places365. These weights had been fine-tuned with our own dataset. Figure 5 shows the change in the loss value in the process of the model training. To prevent overfitting, we trained the model up to epoch 18.

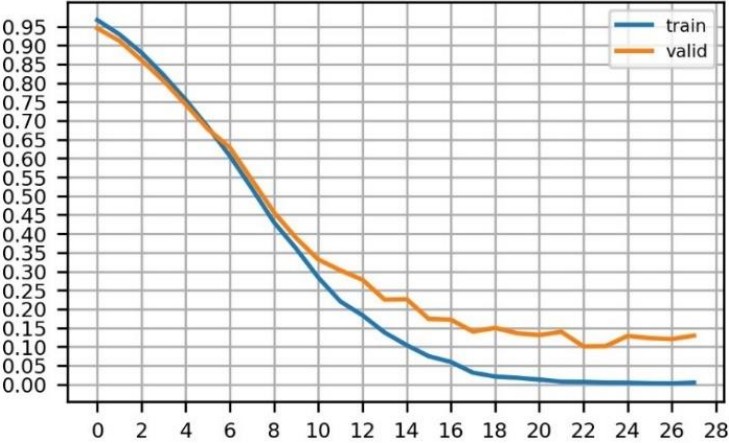

**Figure 5.** Loss graph of model training of Gyeongbokgung Palace.

To remove noise photos, it was necessary to identify the optimal threshold value from the ROC curve. Figure 6a shows the minimum distance between a target photo and the reference set as a histogram. Figure 6b shows the minimum distance between a target photo and the reference set as a histogram. Figure 7 represents the ROC curve, and 0.4 which is

the threshold of the farthest point from Y = X, shown in red X on the graph, was selected as the optimal value. Figure 7 also shows that TPR was 0.928 and FPR was 0.045, which related to the accuracy of the model. Compared with Figure 4b, Figure 8 shows that data points belonging to the same cluster were close to each other and the distances between different clusters were farther apart. Figures 9 and 10 respectively show the number of photos and example photos finally classified by items in a category.

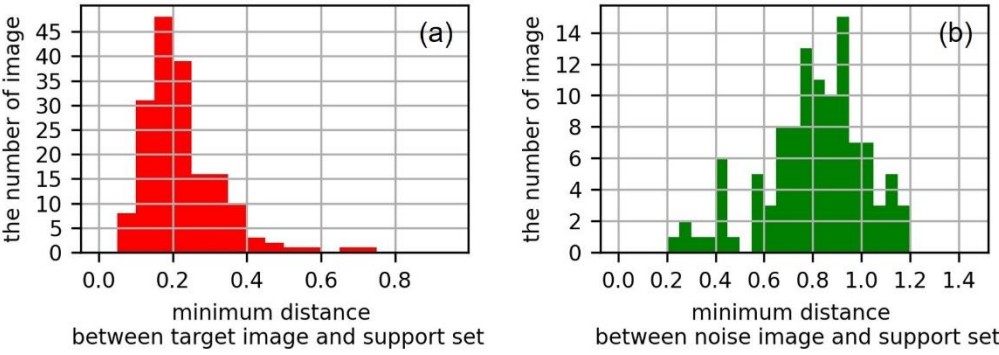

**Figure 6.** Histogram of Gyeongbokgung Palace: (**a**) minimum distance between a target photo set and reference set; (**b**) minimum distance between a noise photo and reference set.

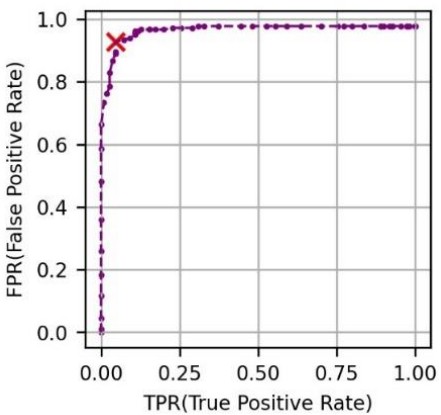

**Figure 7.** ROC curve of Gyeongbokgung Palace.

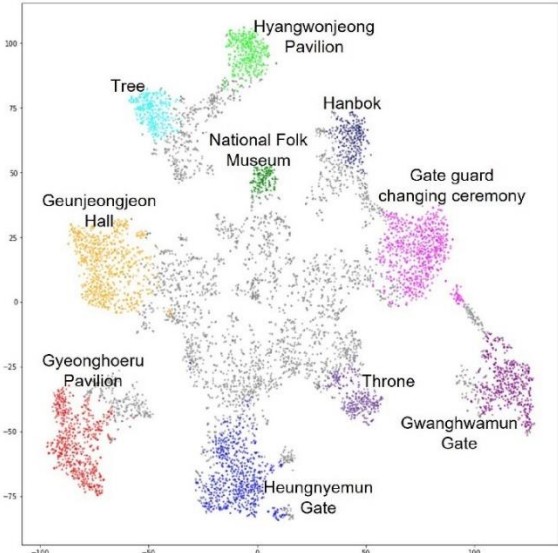

**Figure 8.** Datapoints classified with Siamese Network of Gyeongbokgung Palace.

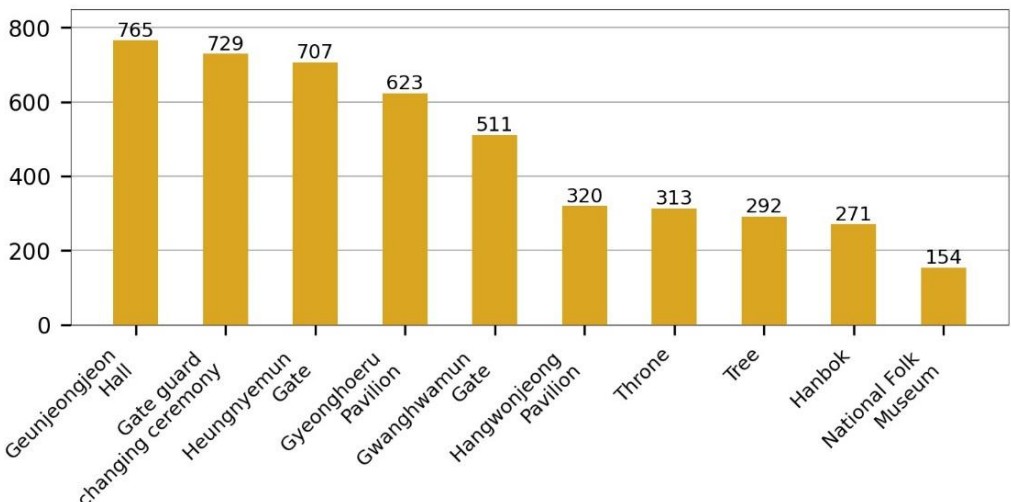

**Figure 9.** Number of photos by category in Gyeongbokgung Palace.

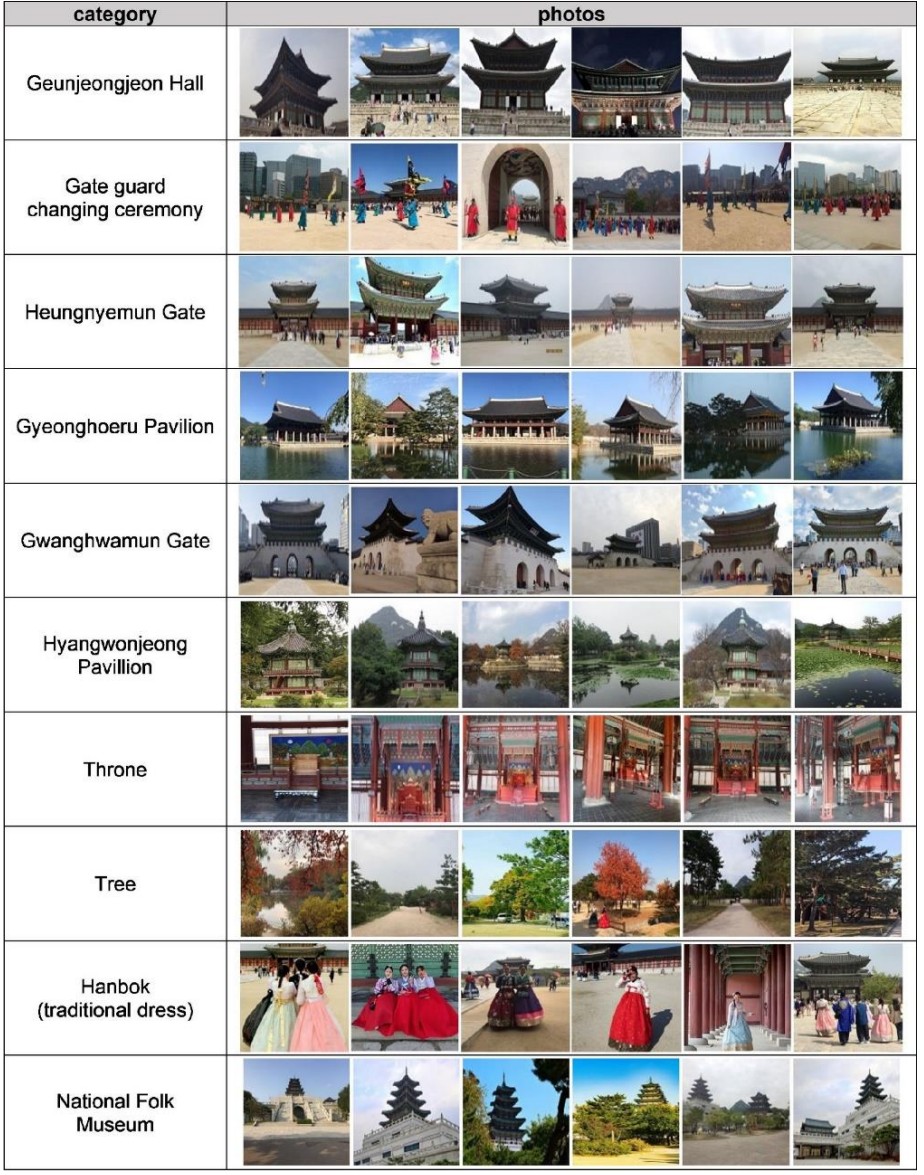

**Figure 10.** Example photos by category in Gyeongbokgung Palace.

### 4.2. Insadong

There are 6410 reviews registered on TripAdvisor's 'Insadong' page. Of these, 3695 photos were collected from 5915 reviews written in a foreign language. Each photo was embedded as a 512-dimensional vector, reduced to two dimensions using t-SNE, and clustered using HDBSCAN. Figure 11 shows the result and Table 3 shows the number of photos for each cluster. Of the total 3659 points, 2568 were classified as noise, and 5 clusters were generated in Figure 11b. Since the 134 photos belonging to cluster 4 represent different visual contents such as signs, souvenirs, murals, portraits, and food, the cluster was not considered in the building category. Four categories were finally created as follows: 'Ssamzigil', 'Insadong street', 'Food and Beverage', and 'Souvenir'.

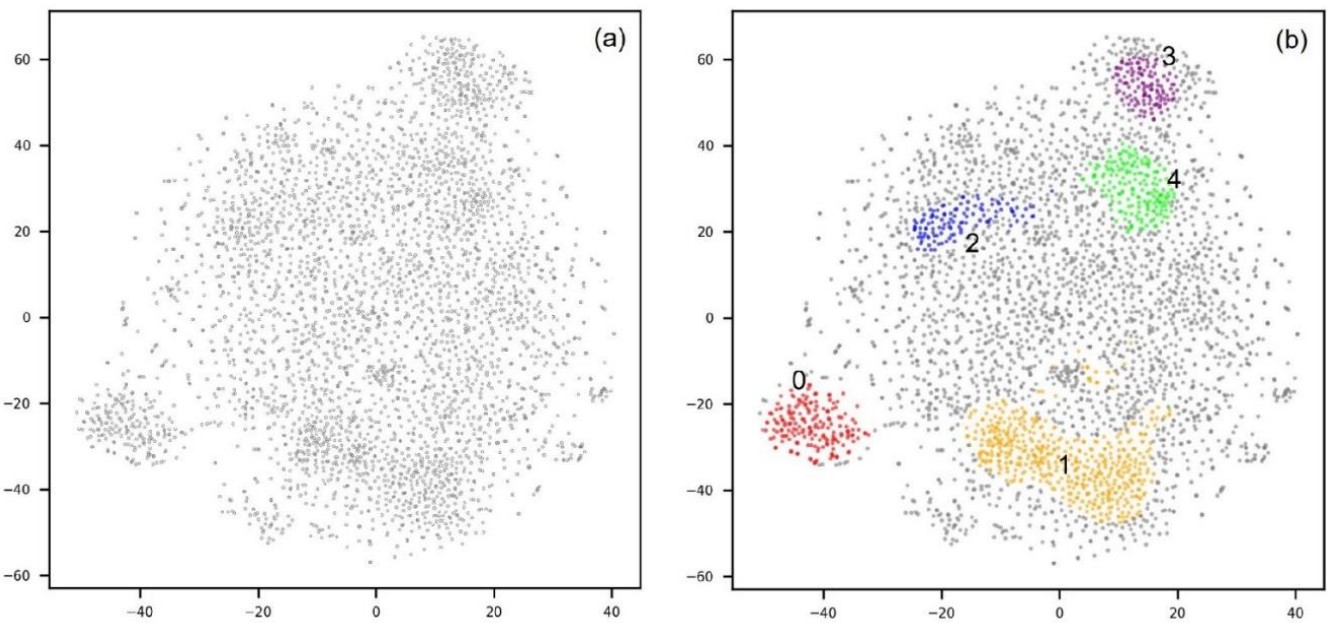

**Figure 11.** Insadong photos: (**a**) two-dimensional visualization; (**b**) result of HDBSCAN.

**Table 3.** Number of photos for each cluster generated as a result of HDBSCAN in Insadong.

| Cluster | Number of Photos |
|---|---|
| 0 | 158 |
| 1 | 478 |
| 2 | 133 |
| 3 | 168 |
| 4 | 134 |

Siamese network was used to classify photos according to the previously generated categories and remove noise photos. For this purpose, we trained the Siamese network on Insadong's photo dataset. At this time, the training dataset was organized using the photos composed of each cluster except for those which were incorrectly included in clusters. Table 4 shows the number of photos included in the training dataset for each category. The model used in 'Insadong' was also based on the VGG16 network. However, unlike the model used in 'Gyeongbokgung Palace', we fine-tuned the top four layers of the model without additional convolutional layers. Since 'Insadong' had a smaller training dataset than 'Gyeongbokgung Palace', overfitting may occur. Figure 12 shows the loss value change during model training, and the model trained up to epoch 10 was used to prevent overfitting.

**Table 4.** Number of training photos by category in Insadong.

| Category | Number of Training Photos |
|---|---|
| Ssamzigil | 150 |
| Insadong Street | 126 |
| Food and Beverage | 132 |
| Souvenir | 137 |

**Figure 12.** Loss graph of model training of Insadong.

To remove the noise photos included in the category, we examined the optimal threshold value through the ROC curve. Figure 13a shows the minimum distance between the test set and the reference set as a histogram. Figure 13b shows the minimum distance between the noise set and the reference set as a histogram. Figure 14 shows the ROC curve, and 0.32, the threshold value of the point farthest from Y = X, corresponding to the red X on the graph, was selected as the optimal value. Figure 14 also shows that TPR was 0.90 and FPR was 0.057, which related to the accuracy of the model. Figure 15 shows the trained model and data points classified by threshold. Compared with Figure 11b, Figure 15 shows that data points belonging to the same cluster were close to each other, and the distances between different clusters were farther apart. Figures 16 and 17 respectively show the number of photos and example photos finally classified by category.

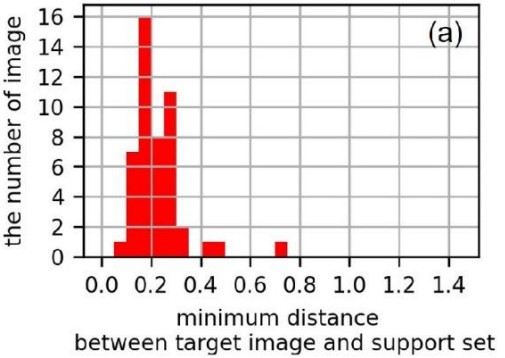 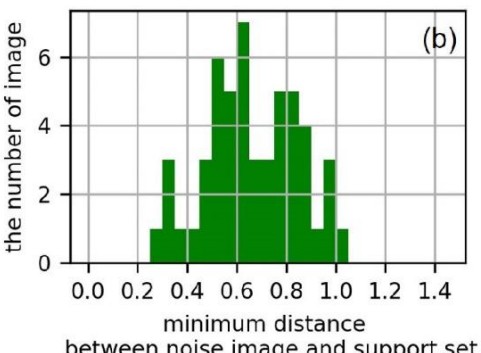

**Figure 13.** Histogram of Insadong: (**a**) minimum distance between a target photo set and reference set; (**b**) minimum distance between a noise photo and reference set.

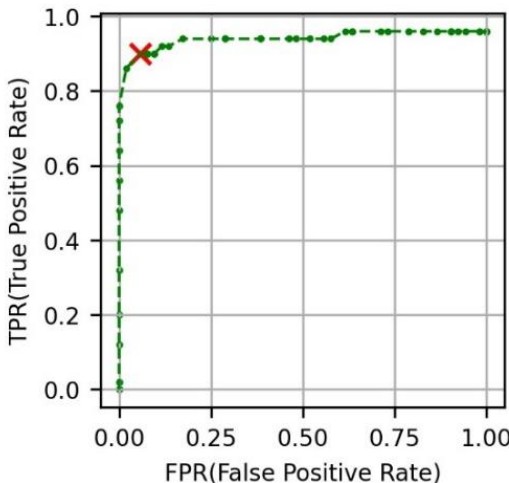

**Figure 14.** ROC curve of Insadong.

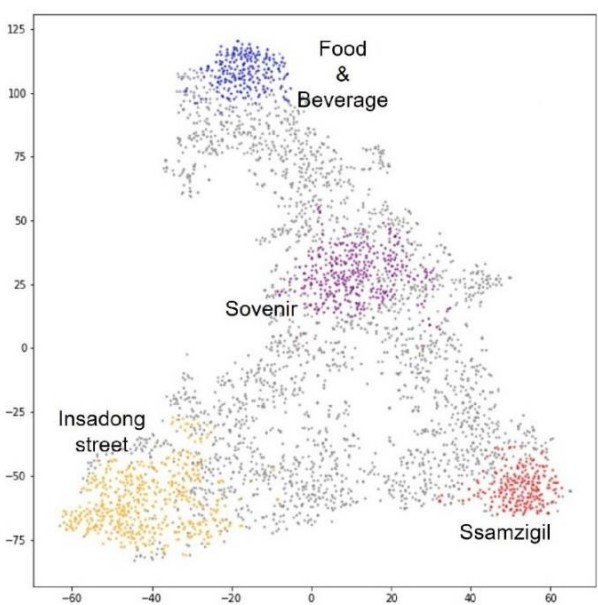

**Figure 15.** Datapoints classified with Siamese Network of Insadong.

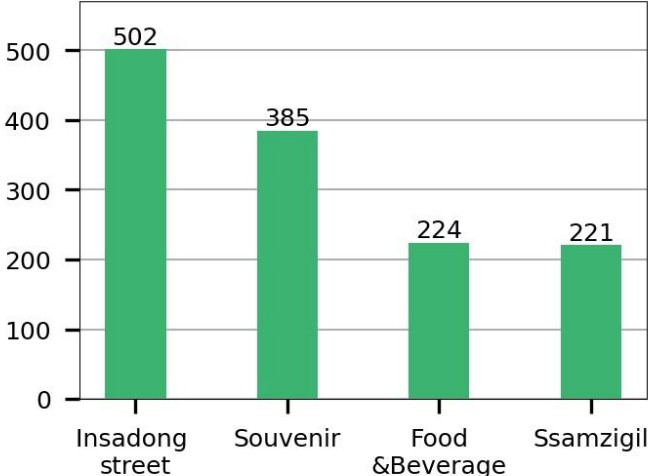

**Figure 16.** Number of photos by category in Insadong.

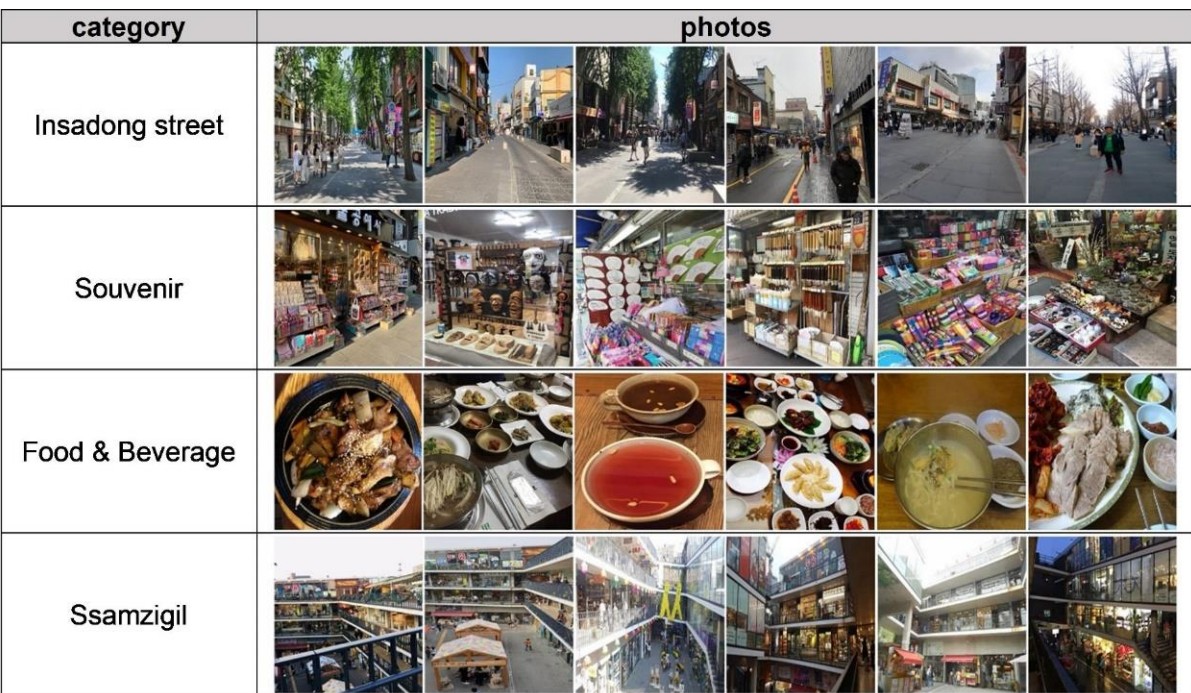

**Figure 17.** Example photos by category in Insadong.

## 5. Discussion and Conclusions

As the value of photos posted by tourists is recognized as more and more important in the tourism field, new approaches to analyzing tourist photos using deep learning technology are being attempted. The research methods that analyze tourism photos using recent deep learning technology are two-fold. The first method is that tourism images are analyzed after classifying tourist photos by predetermined photos classification categories such as Places365 or ImageNet. The second method is that tourism images are analyzed according to a tourism photo classification category generated on a city or national scale. In the former case, there is a shortcoming in that unique photos appearing in specific tourist attractions cannot be properly classified with a category designed for general purposes. In the latter case, there are limitations in that it puts a lot of time and effort into building a category and dataset and has difficulty detecting the locality of tourism attractions.

The purpose of this study is to propose a method for automatically building a category for each attraction by clustering photos and classifying them with a Siamese network, rather than classifying them into predetermined categories. In addition, this study attempts to confirm the validity of the proposed method by applying it to two representative tourist attractions in Seoul. This study has four steps to clarify the photo classification method for each tourist attraction and to confirm its validity. First, we collected tourist photos attached to reviews posted by foreign tourists on TripAdvisor. Second, we embedded photos as feature vectors in 512 dimensions using the VGG16 network pre-trained with Places365 and reduced them to 2 dimensions using t-SNE. Third, to create a category based on visual contents that frequently appear in photos taken by tourists, clusters were extracted through HDBSCAN analysis and they were set as an image category of an attraction. Fourth, we removed the noises in the cluster through the Siamese network and analyzed the image of tourist attractions by confirming the number of classified photos in each category.

Using the method proposed in this study, the Tripadvisor photos posted by foreign tourists in 'Gyeongbokgung Palace' and 'Insadong' in Seoul, Korea were analyzed. Gyeongbokgung Palace is a palace built during the Joseon Dynasty and is one of the representative tourist attractions located in the downtown area of Seoul. In Gyeongbokgung Palace, 10 categories were created as follows: 'Geunjeongjeon Hall', 'Gyeonghoeru Pavilion', 'Heungnyemun Gate', 'Hyangwonjeong', 'National Folk Museum', 'Throne', 'Hanbok(Korean

traditional dress)', 'Gate guard changing ceremony' and 'Tree' 'Gwanghwamun Gate'. Through this, it was possible to check which destination images of 'Gyeongbokgung Palace' are preferred by foreign tourists. 'Insadong' is also one of the representative tourist attractions in the downtown area of Seoul. Insadong is an area known as an exhibition center for Korean traditional arts, antiques, and old ceramics that have been handed down generation after generation. In Insadong, four categories were created: 'Ssamzigil', 'Insadong street', 'Food and Beverage', and 'Souvenir'. Through this, it was possible to identify which images of Insadong are preferred by foreign tourists.

This study is differentiated from the existing studies in the following three aspects. First, since we make categories based on clustering results, features that make tourism destinations attractive can be identified more specifically and flexibly in a data-driven manner. Second, since we set the results of clustering analysis as categories, it is not necessary to manually build the training dataset. Third, to address the scarcity of data, we employ a Siamese network that can improve classification performance with a rather small volume dataset. In the case of the tourism field, if the research area is narrowed down to a specific tourist attraction, there may be a limit to the amount of data that can be used. However, since the data used in this study is the photos posted on TripAdvisor, there is a possibility that various photos may be less mixed because the categories are divided into tourist destinations, tourist attractions, and activities. Therefore, it is necessary to compare the photos posted on TripAdvisor with those from other SNS sites that are likely to post various photos for the same area. In addition, it is necessary to compare a category created by the proposed method with one by an existing method proposed by previous studies, which classifies tourist photos by predefined photo categories using Places365 or ImageNet.

**Author Contributions:** Conceptualization, Jiyeon Kim and Youngok Kang; Methodology, Software, Validation, Formal Analysis, Jiyeon Kim; Writing-Original Draft Preparation, Jiyeon Kim; Writing-Review & Editing, Youngok Kang; Supervision, Youngok Kang; Project Administration, Youngok Kang; Funding Acquisition, Jiyeon Kim. All authors have read and agreed to the published version of the manuscript.

**Funding:** This research was supported a grant from geospatial information workforce development program funded by the Ministry of Land, Infrastructure and Transport of Korean Government (1 March 2020).

**Institutional Review Board Statement:** Not applicable.

**Informed Consent Statement:** Not applicable.

**Data Availability Statement:** Not applicable.

**Conflicts of Interest:** The authors declare no conflict of interest.

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
