# Peer review of "Automatic Classification of Photos by Tourist Attractions Using Deep Learning Model and Image Feature Vector Clustering"

_ijgi, doi:10.3390/ijgi11040245_

Round 1

Reviewer 1 Report

Very interesting topic - applicable in real-world practice.

The scientific quality of the manuscript is excellent! 

There should only be one minor correction - authors are adviced to explain the abbreviations at their first occurence in text, starting with abstract, where there are VGG16, HDBSCAN, and later in introduction SNS...

Author Response

  • Thank you for your review.
  • After revision, we have explained abbreviations at their first occurrence as follows:  ‘user generated contents (UGC)’ in line 8, ‘tourism destination images (TDI)’ in line 9, ‘t-SNE(t-Distributed Stochastic Neighbor Embedding)’ in line 21, ‘HDBSCAN (Hierarchical Clustering and Density-Based Spatial Clustering of Applications with Noise)’ in line 22, social network service (SNS) in line 43 and so forth.

Reviewer 2 Report

The paper discusses a novel approach to identify features of interest in tourist attraction areas by analysing and clustering photos extracted from Trip Advisor.

I believe this is a very interesting work, which rather than classifying pictures with pre-trained models, implement clustering to obtain classes of images in a data-driven fashion and discover new knowledge on what specific features make places especially attractive.

While I like the work I also think it is not very well presented. I had to read the results section to really appreciate the authors ideas and ambitions. Therefore, I would suggest the following revisions on the text:

*text needs to be substantially proof read

* the research problem is poorly framed. The introduction is very short and it does not give a clear sense of what the contribution is and what the research aims are.

* when the authors say that they classify tourist photos by tourist attractions it suggests that you want to detect tourist attractions, but it is not what they do. The data collection targets very specific tourist attractions first and then categorise features of tourist interest through clustering them. The overall process the authors engaged with and the research aims need to be clarified up front insofar readers can imagine what kind of output the approach will generate before going through the results.

* I would discuss the role of clustering more extensively as it is pivotal to move from a set of pre-determined categories to data-driven knowledge discovery

Author Response

  • We are really appreciated your concrete advice. They are greatly helpful to developing our study.
  • We have corrected the awkward sentences and expressions in entire text to increase readability and to convey the intended meaning.
  • To clarify our contributions and purpose of this study, we have highlighted the differences from existing studies in line 87 ~98 , to shape up our research process, we have added sentences to research process part in introduction in line 91~98, and added the related contents in detail in line 60 ~ 91.
  • We have stated the role of clustering method in detail in line 242~246.

Reviewer 3 Report

Manuscript analyses tourism photos using recent deep learning technology. The manuscript is clear and well structured. Below, the authors can find some minor revisions.

Line 124           SIFT, SUFF, and BRIFF … specify acronyms

Line 129           please revise the sentence because it is not clear

Line 149           please revise the sentence because it is not clear

Line 242           please add a reference

Line 486           Please review the reference

Author Response

  • Thank you for your review.
  • Line 124 : SIFT, SUFF, and BRIFF … specify acronyms.   

    -> We have specified those abbreviations in line 143~145.

  • Line 129 : please revise the sentence because it is not clear.

    -> We have revised the sentence in line 149~153.

  • Line 149: please revise the sentence because it is not clear.

    -> We have revised the sentence in line 166~169.

  • Line 242: please add a reference.

    -> We have added reference [36] in line 258.

  • Line 486 Please review the reference.

    -> We have reviewed the reference in line 494. and corrected author's name